# Investigation of High Frequency Irreversible Electroporation for Canine Spontaneous Primary Lung Tumor Ablation

**DOI:** 10.3390/biomedicines12092038

**Published:** 2024-09-07

**Authors:** Alayna N. Hay, Kenneth N. Aycock, Melvin F. Lorenzo, Kailee David, Sheryl Coutermarsh-Ott, Zaid Salameh, Sabrina N. Campelo, Julio P. Arroyo, Brittany Ciepluch, Gregory Daniel, Rafael V. Davalos, Joanne Tuohy

**Affiliations:** 1Department of Small Animal Clinical Sciences, Virginia Maryland College of Veterinary Medicine, Blacksburg, VA 24061, USA; anw5137@vt.edu (A.N.H.); gdaniel@vt.edu (G.D.); 2Virginia Tech Animal Cancer Care and Research Center, Virginia Maryland College of Veterinary Medicine, Roanoke, VA 24016, USA; 3Department of Biomedical Engineering and Mechanics, Virginia Polytechnic Institute and State University, Blacksburg, VA 24061, USAlorenzomelvin76@gmail.com (M.F.L.); kdavid39@gatech.edu (K.D.); zss@gatech.edu (Z.S.);; 4Wallace H. Coulter Department of Biomedical Engineering, Georgia Institute of Technology and Emory University, Atlanta, GA 30318, USA; 5Department of Biomedical Sciences and Pathobiology, Virginia Maryland College of Veterinary Medicine, Blacksburg, VA 24061, USA; slc2003@vt.edu

**Keywords:** tumor ablation, lung cancer, comparative oncology, canine oncology

## Abstract

In this study, the feasibility of treating canine primary lung tumors with high-frequency irreversible electroporation (H-FIRE) was investigated as a novel lung cancer treatment option. H-FIRE is a minimally invasive tissue ablation modality that delivers bipolar pulsed electric fields to targeted cells, generating nanopores in cell membranes and rendering targeted cells nonviable. In the current study, canine patients (*n* = 5) with primary lung tumors underwent H-FIRE treatment with an applied voltage of 2250 V using a 2-5-2 µs H-FIRE waveform to achieve partial tumor ablation prior to the surgical resection of the primary tumor. Surgically resected tumor samples were evaluated histologically for tumor ablation, and with immunohistochemical (IHC) staining to identify cell death (activated caspase-3) and macrophages (IBA-1, CD206, and iNOS). Changes in immunity and inflammatory gene signatures were also evaluated in tumor samples. H-FIRE ablation was evident by the microscopic observation of discrete foci of acute hemorrhage and necrosis, and in a subset of tumors (*n* = 2), we observed a greater intensity of cleaved caspase-3 staining in tumor cells within treated tumor regions compared to adjacent untreated tumor tissue. At the study evaluation timepoint of 2 h post H-FIRE, we observed differential gene expression changes in the genes *IDO1*, *IL6*, *TNF*, *CD209*, and *FOXP3* in treated tumor regions relative to paired untreated tumor regions. Additionally, we preliminarily evaluated the technical feasibility of delivering H-FIRE percutaneously under CT guidance to canine lung tumor patients (*n* = 2). Overall, H-FIRE treatment was well tolerated with no adverse clinical events, and our results suggest H-FIRE potentially altered the tumor immune microenvironment.

## 1. Introduction

In both veterinary and human patients, primary lung cancer does not have favorable long-term survival outcomes, and an advancement in treatment options is profoundly needed to improve prognosis. Primary lung cancer accounts for approximately 1% of canine neoplasia with incidences as high as 8.8% in an evaluated canine population in the United Kingdom [1,2]. Human primary lung cancer is the second most common cancer and leading cause of cancer-related deaths in the United States, in both men and women [3,4]. Lung adenocarcinoma is the predominantly occurring primary lung cancer in both canines [5] and humans [4].

While the utility of canine lung cancer as a comparative model for human disease is not as well characterized compared to cancers like osteosarcoma [6,7,8], there are shared similarities between the two species. These similarities include mutations in the oncogene *K-ras* [9] and biologic similarities such as the prognostic significance of metastatic disease in both species [5,10,11]. The current average survival time for canine lung cancer patients who have undergone the surgical resection of a primary tumor is 10–18 months [5,12]. The 5-year survival rate for human patients who undergo standard of care treatment, and considering all stages of lung cancer, is a grim 18% [3,4]. The treatment of primary lung tumors is multimodal, but surgical resection remains the mainstay treatment for the primary tumor. However, not all lung cancer patients are appropriate surgical candidates due to factors such as co-morbidities, diffuse lesions within multiple lung lobes, and/or tumor location. Additional primary tumor treatment modalities include radiation, chemotherapy, immunotherapy, and more recently, thermal ablative techniques such as radiofrequency and microwave ablation, which have their limitations [13,14,15,16]. Potentially the greatest limitation of thermal ablation modalities is the heat sink effect, which limits their use near critical structures such as blood vessels [17,18,19]. The poor prognosis of lung cancer patients, combined with the potential limitations and the invasive nature of surgery, necessitate innovative avenues of therapeutic investigation to improve treatments and outcomes for primary lung tumors.

Irreversible electroporation (IRE), a minimally thermal ablation technique which delivers monopolar and monophasic electrical pulses to targeted tissue [20], has been explored as a minimally invasive ablation modality for lung tumors [21,22,23,24,25,26,27]. The technique is characterized as minimally thermal due to the temperature changes which occur in the tissue directly surrounding the electroporation probes which may lead to minimal thermal damage [28]. However, the predominant mechanism of cell death is non-thermal [29,30]. The feasibility and short-term safety of delivering IRE to porcine lung tissue has previously been reported [21,25,26]. The feasibility of delivering IRE to human lung tumors was previously investigated [24]. To date, there has been one human clinical trial that utilized IRE for the treatment of lung cancer, and limited success was reported, likely due to the differences in electrical conductivity between the pulmonary parenchyma and tumor tissue [27]. IRE also has limitations as a tumor ablation modality such as the induction of cardiac arrhythmias and muscle contractions, necessitating the use of paralytics and cardiac synchronization during the procedure [31]. The modified version of IRE, high-frequency irreversible electroporation (H-FIRE), has potential to capitalize on the advantages of traditional IRE while overcoming these limitations.

H-FIRE utilizes bipolar, microsecond length pulses (0.5–10 µs duration) of electrical current discharged between paired electrodes to induce pore formation in cell membranes that disrupt molecular transport and ultimately trigger cell death [32,33]. The delivery of bipolar instead of monopolar pulses, as in traditional IRE, is advantageous because it negates challenges such as severe muscle contraction and cardiac electrical asynchrony leading to cardiac arrhythmias [34,35,36]. H-FIRE mitigates muscle contractions due to the short pulse lengths and rapid change in the polarity of the pulses which limits nerve or muscle stimulation [37,38]. Thus, H-FIRE, with its advantages over IRE, represents an innovative, minimally invasive, and minimally thermal ablation technique that has potential to ablate lung tumors and help patients avoid the morbidities of surgical tumor resection. Furthermore, our group has successfully demonstrated the feasibility of employing H-FIRE for the partial ablation of spontaneously occurring canine liver tumors [35], canine intracranial meningioma [39], and superficial tumors in equines [40,41,42].

Not only is H-FIRE capable of ablating macroscopic tumors, but it has also demonstrated immunomodulatory effects that lead to immune activation against tumors, thus raising the exciting potential to mitigate metastatic disease development. While the exact mechanism of H-FIRE induced immune activation remains to be elucidated, it is reported that H-FIRE ablation can elicit the release of tumor-associated antigens and damage-associated molecular patterns (DAMPs), all of which can serve to activate the immune system to recognize and eliminate tumor cells [35,43,44,45,46,47,48,49]. H-FIRE, with its ability to ablate solid tumors and its potential for immunogenic stimulation, is uniquely poised to achieve the dual goals of lung tumor therapy—eliminating the primary tumor and mitigating metastatic disease development. Prior to investigations by our group [50], H-FIRE has not been reported for lung tumor ablation. Given the exciting potential to utilize H-FIRE as a minimally invasive lung tumor ablation technique, we conducted the following studies to develop H-FIRE for this application: (i) a proof-of-principle clinical study to evaluate the ablative outcomes, safety, and feasibility of delivering H-FIRE ablation in spontaneously occurring canine pulmonary tumors in client-owned dogs and (ii) an in vitro study to assess cell death and immunomodulatory responses of canine lung tumor cells to H-FIRE ablation.

## 2. Materials and Methods

### 2.1. Canine Clinical Study

#### 2.1.1. Patient Selection

Client-owned dogs with primary lung tumors (*n* = 7) were recruited for the clinical study under an approved Virginia Tech Institutional Animal Care and Use Committee protocol (IACUC protocol number 20-072) and owner consent was obtained. The criteria for clinical study enrollment were (1) a radiographic diagnosis of a pulmonary mass suspected to be a primary lung tumor, (2) the owners have elected for a surgical resection of their dog’s lung tumor, and (3) there was no previous tumor-directed therapy prior to enrollment.

#### 2.1.2. Treatment Planning

Pre- and post-contrast computed tomography (CT) scans of the thorax using a Siemens Somatom Confidence RT scanner were performed for enrolled patients for H-FIRE ablation and surgical planning. A semi-ellipsoidal volume with approximate dimensions of 3 cm (h) × 2 cm (w) × 1.5 cm (d) was planned for H-FIRE treatment in a portion of the tumor, leaving the remainder of the tumor untreated for comparative evaluation. An open-source medical image processing software (3D Slicer v4.11) was used to evaluate and segment tumors and pertinent peri-tumoral anatomic structures within the proximity of the tumor such as major vasculature that was visualized on CT scans. After segmentation and interpolation between imaging planes, segmented structures were imported into a medical 3D printing software (Mimics Innovation Suite, Materialise, Leuven, Belgium, v24) for smoothing and wrapping, creating a contiguous 3D structure. For calculating estimated tumor ablation volumes and computing the electrical potential to apply during H-FIRE tumor ablation, these structures were imported into a finite element analysis software (Comsol Multiphysics^®^ v5.5, Comsol Inc., Burlington, MA, USA) for the subsequent computation of electric potential and field distribution as previously described [35]. Specific details regarding Comsol outcomes for the intra-operative H-FIRE cases (*n* = 5) reported in this manuscript have previously been reported [50].

#### 2.1.3. Delivery of H-FIRE Ablation

##### Delivery of Intra-Operative H-FIRE Treatment

H-FIRE ablation occurred with patients under general anesthesia in the operating suite, prior to the standard-of-care surgical resection of the primary lung tumors (*n* = 5). A custom built H-FIRE generator capable of producing microsecond-duration bipolar pulses in rapid bursts was utilized to deliver H-FIRE therapy via a pair of 18-gauge bipolar electrodes (AngioDynamics Inc., Latham, NY, USA) (Figure 1A). The spacing between the electrodes on each probe was set so that the 7 mm distal electrode was spaced a distance of 8mm away from the 8 mm proximal electrode. The spacing between the bipolar probes varied between 1–1.5 cm depending on tumor size. Probes were inserted into a custom-made holder to keep probes parallel to one another, then the probe tips were inserted approximately 4 cm into the tumor during open surgery (Figure 1B). Prior to treatment delivery, a sequence of 3 bursts was delivered to the tumor at a sub-electroporative level (25 V) and then incrementally increased to the treatment voltage of 2250 V to obtain tissue-specific conductivity data as previously described [50]. For treatment, the generator was set to deliver sets of 25 bursts across each of the four electrode pairs until 100 total bursts per pair were delivered (400 bursts total). A 30 s delay was implemented after every other set of 25 bursts to minimize the potential for the treatment to produce thermal damage. Electrical signals were monitored with a high-voltage differential probe (Harvard Apparatus, BTX Enhancer 3000, Holliston, MA, USA) and a passive current probe (Pearson Electronics, #2877, Palo alto, CA, USA) connected to an oscilloscope (TeleDyne LeCroy, Wavesurfer 3024z, Chestnut Ridge, NY, USA). H-FIRE was delivered with the dogs under inhalation general anesthesia without intraoperative paralytic agents. The dogs’ physiologic parameters including heart rate, blood pressure, and electrocardiography (ECG) readings were monitored for any abnormalities throughout the duration of H-FIRE treatment and subsequent surgery.

##### Delivery of CT-Guided Percutaneous H-FIRE Treatment

Upon verifying the absence of adverse events with delivering intra-operative H-FIRE treatment to pulmonary tumors, we preliminarily evaluated the safety and feasibility of delivering CT-guided percutaneous H-FIRE ablation to pulmonary tumors in canine patients (*n* = 2). These preliminary treatments aimed to advance our goal of developing H-FIRE as a minimally invasive tumor ablation modality. Prior to delivering percutaneous H-FIRE treatment, patients received a thoracic CT scan for H-FIRE ablation and surgical planning as described above (see treatment planning section). The patients were maintained under general anesthesia, and a single 18-gauge bipolar H-FIRE electrode was inserted percutaneously under CT guidance by a board-certified veterinary radiologist (G.D.). After CT confirmation of appropriate probe placement, the probe was held in place by a custom-designed probe positioner, and a sequence of 3 H-FIRE bursts was delivered at increasing voltage amplitudes to determine the maximum operating voltage. The maximum operating voltage was established before consistent muscle contractions were observed or when the target voltage of 2250 V was reached, whichever occurred first. The target voltage was determined in the pre-operative treatment planning to achieve an ablation volume of 3 cm^3^ with the 2-5-2 µs (positive phase–interphase delay–negative phase) H-FIRE waveform. This waveform cycle was repeated 25 × per burst for a burst energized time of 100 µs. Following the pre-treatment voltage ramp, H-FIRE ablation was delivered to a portion of the tumor using the custom built generator. The single-needle treatment consisted of 200 bursts delivered in 4 sets of 50 bursts each, with 30 s delays between sets to mitigate thermal damage. The bursts were delivered during the refractory period of the heart by utilizing a cardiac trigger monitor (AccuSync 72 ECG, AccuSync, Milford, CT, UK) to trigger H-FIRE bursts. Treatment was monitored with the same equipment as the intra-operative procedures. Immediately after the completion of H-FIRE ablation, a thoracic CT scan was repeated to identify signs of pneumothorax and hemorrhage.

#### 2.1.4. Tissue Conductivity Assessment

To obtain tissue-specific conductivity data, an in-situ voltage ramp was employed immediately prior to the administration of H-FIRE treatment, as described above. Current and voltage data were recorded on an oscilloscope (Teledyne LeCroy, Chestnut Ridge, NY, USA) for each pre-treatment voltage ramp sequence, capturing 3 H-FIRE bursts for each voltage (25, 50, 250, 500, 1000, 1500, 2000, and 2250 V) or increasing until the maximum operating voltage was reached. We replicated the treatment geometry in a 3D finite element model to simulate the current respective to each voltage. In the numerical model, the tissue conductivity was defined to be dependent on electric field, related by a sigmoidal function. The fitting parameters for this relationship were iteratively modified to minimize the root mean square error between the current obtained from the numerical model and the experimental data as described in [43,50,51].

#### 2.1.5. Surgical Tumor Resection

After intra-operative or percutaneous H-FIRE ablation, patients remained under general anesthesia, and a standard-of-care surgical resection of lung tumors was performed by board-certified veterinary surgical oncologists (J.T. and B.C.). Tumor resection was performed at least 2 h after the H-FIRE ablation of the tumors. This time-point was selected to allow for the initial ablative effects of H-FIRE to occur [32,47] prior to surgical resection. After the surgical procedure was completed, the patients were recovered from anesthesia and brought to the intensive care unit for post-operative monitoring and care. Patients were discharged to the care of their owners when deemed appropriate by the attending clinician.

#### 2.1.6. Adverse Event Reporting

Patients were monitored for any adverse events intraoperatively and post-operatively. Any adverse events associated with H-FIRE lung tumor ablation were graded using the Veterinary Cooperative Oncology Group Common Terminology Criteria for Adverse Events (VCOG-CTCAEv2) [36].

#### 2.1.7. Gross and Microscopic Tumor Evaluation

Following surgical resection, the primary tumor was grossly and microscopically evaluated by a board-certified veterinary pathologist (S.C.-O.) to identify the ablation zone and microscopic characteristics of H-FIRE ablation. For the gross identification of the H-FIRE ablation site, the project principal investigator (J.T.) placed sutures to mark the H-FIRE probe insertion sites (Figure 2A) and photographs were taken during the ablation procedure to demonstrate the direction and depth of H-FIRE electrode insertion. All collected tumors were sectioned serially parallel to the long axis of the H-FIRE electrode probes to ensure consistency for evaluation. After gross evaluation, treated and untreated portions of the tumor were collected and fixed in 10% neutral buffered formalin and paraffin embedded (FFPE). The untreated tumor samples were collected as far away as possible from the electrode probe insertion site. For the microscopic evaluation of H-FIRE ablation, 5 μm treated and untreated FFPE tumor tissue samples were sectioned and stained with hematoxylin and eosin using an automated stainer, Ventana Discovery Ultra Automated Research Stainer (Roche Diagnostics, Indianapolis, IN, USA). Tumor samples were evaluated for the demarcation of the ablated area, tumor cell death, intact tumor cells, and hemorrhage.

#### 2.1.8. Immunohistochemistry (IHC)

Chromogenic multiplex and single IHC staining were completed to identify macrophages and to evaluate apoptotic cell death with activated caspase-3 staining, respectively. For all IHC staining, paired H-FIRE-treated and untreated tumor tissue sections (5 µm) were stained for comparative evaluation for each patient that received intraoperative H-FIRE treatment. IHC staining was conducted with a Roche Ventana Discovery Ultra Automated Research Stainer (Roche Diagnostics, Indianapolis, IN, USA). For multiplex IHC macrophage phenotyping, the panel consisted of IBA1 (FujiFilm, 019-19741,1:300, Tokyo, Japan), iNOS (Abcam, ab3523, 1:100, Cambridge, UK), and CD206 (NovusBio, NBP190020, 1:200, Centennial, CO, USA). For activated caspase-3 staining, rabbit polyclonal anti-active caspase-3 antibody (Promega, Madison, WI, USA) was used at a 1:200 dilution. All sections were qualitatively evaluated by a board-certified veterinary pathologist (S.C.-O.) for positive staining of the listed markers in treated and untreated tumor sections. The IHC staining was only completed with the tumor samples treated with H-FIRE intra-operatively and not the percutaneously treated samples due to the limited sample size.

#### 2.1.9. Gene Expression Arrays

For the patients who received intra-operative H-FIRE treatment (*n* = 5), differential gene expression analysis was conducted. Total RNA was extracted from 2, 20 μm FFPE tissue scrolls sectioned from untreated and treated regions of all patient tumor samples with the Zymo Quick-RNA FFPE extraction kit (Zymo, Irvine, CA, USA). The manufacturer’s protocol was followed (Zymo, Irvine, CA, USA). RNA was assessed with a NanoDrop One C (Thermo Fisher, Waltham, MA, USA), and samples had a 260/280 ratio of 1.8–2.0 (*n* = 5 paired patient samples). The Qiagen RT^2^ First Strand kit was utilized to synthesize 400 ng RNA to cDNA in accordance with the manufacturer’s protocol (Qiagen, Hilden, Germany). A custom 96 well RT^2^ PCR canine specific Cancer Immunity and Inflammation Crosstalk gene array (Qiagen, Hilden, Germany) was utilized for differential gene expression analysis. Each array contained 89 genes relating to inflammation, immunity, and cancer, 3 housekeeping genes, and 4 internal quality control samples to evaluate genomic DNA contamination, reverse transcriptase, and assay efficiency (see Appendix A for full gene list). Arrays were analyzed with an Applied Biosystems 7500 Fast Real-Time PCR System (Thermo Fisher Waltham, MA, USA). The arithmetic mean of the house keeping genes HPRT1, GAPDH, and ACTB was used for normalization, and gene expression changes in treated samples were calculated relative to paired untreated regions of the tumor with the standard ΔΔCt methodology. The differential gene expression analysis was only completed with the tumor samples treated with H-FIRE intra-operatively and not the percutaneously treated samples due to the limited sample size. One gene array was completed per patient, as directed by the manufacturer (i.e., technical replicates were not deemed necessary).

### 2.2. In Vitro Study

It has previously been reported that electrical pulsed-field-based cell death occurs through various mechanisms and is not fully complete until 12–24 h after treatment [32,52]. Thus, given the treat-and-resect nature of the canine clinical study, definitive conclusions regarding the size of the complete ablated volumes were not possible. In an effort to fill this knowledge gap, we conducted in vitro experiments to examine the ablative capacity of H-FIRE on canine lung adenocarcinoma cells (CLAC) in 3D culture conditions.

#### 2.2.1. In Vitro H-FIRE Ablation of 3D Hydrogel Cell Culture Scaffold for Cell Death Assessment

##### Cell Culture

The CLAC cell line (JCRB cell bank, Sekisui XenoTech, LLC, Kansas City, MO, USA) was cultured in Dulbecco’s Modified Eagle Medium (DMEM, ATCC #30-2002, Manassas, VA, USA) supplemented with 10% fetal bovine serum (R&D Systems, #S11550H, Minneapolis, MN, USA), 1% penicillin–streptomycin (Thermo Fisher, Waltham, MA, USA, product #15140122), and 1× non-essential amino acids (Thermo Fisher, Waltham, MA, USA, product #11140) according to the supplier’s protocol. Cells were cultured at 37 °C with 5% CO_2_ and passaged at 80–90% confluency.

##### Collagen Scaffold Fabrication

The walls of a 12-well culture plate were coated in vacuum grease (Dow Corning, Midland, MI, USA) to prevent hydrogels from forming a meniscus. Concentrated collagen stock solutions were prepared using rat tail collagen type I extracted in-house as previously described [53]. Collagen was neutralized with 10% (*v*/*v*) 10 × DMEM and 1 N NaOH was titrated to achieve a pH of 7–7.4. Cells in suspension were mixed into the solution at 1 × 10^6^ cells/mL to achieve a final collagen concentration of 5 mg/mL. The solution was carefully dispensed into a 12-well tissue culture-treated plate with care taken to ensure even coating on the bottom of each well. Collagen scaffolds were incubated at 37 °C to allow for polymerization, covered in fresh media, and incubated until treatment.

##### H-FIRE Treatment and Viability Assessment of Collagen Scaffolds

Approximately 24 h after seeding, media was removed from cell-laden collagen scaffolds. To match the clinical variation seen in patients, burst delivery rates were set to 45, 60, or 90 burst per minute (bpm) and delivered with a custom built bipolar pulse generator. Circular ablations were created using a custom single-needle (⌀ = 1.65 mm) + grounding ring (⌀ = 18 mm) geometry (Appendix A). The H-FIRE protocols mimicked in vivo burst delivery patterns which were based on heart rate for each patient treated with intra-operative H-FIRE. For all H-FIRE protocols, 200 bursts were delivered; waveforms were composed of a 2 µs pulse duration and 5 µs interphase/interpulse delays (2-5-2 waveform) and were applied at 500 V with 100 μs of energized time per burst.

For determining lesion area, the viability of the scaffolds was assessed. Scaffolds were incubated at 37 °C with 5% CO_2_ for 24 h after H-FIRE treatment, media was then aspirated, and scaffolds were stained with 2.5 µm calcein AM (ThermoFisher, Waltham, MA, USA, product #C34852) and 22 µm propidium iodide (ThermoFisher, Waltham, MA, USA, product #P3566) in 1 × PBS (Corning, Corning, New York, NY, USA, product #21-040-CV) to visualize live and dead cells, respectively. Lesion area and diameter were measured using ImageJ version 1.53n (National Institutes of Health), then related to the numerically computed electric field distribution as previously described [50,54]. For correlating in vitro caspase-3 activation post H-FIRE treatment to in vivo patient tumor sample observations, scaffolds were fixed in 10% formalin and paraffin embedded, sectioned into 5 µm sections, and stained for active caspase-3 as described above (see Section 2.1.8).

#### 2.2.2. Immune Evaluation of CLAC Cells Treated In Vitro with H-FIRE

The CLAC cell line was cultured, as described above, in 2D cell culture plates. For in vitro immune evaluation experiments, passage 7–10 cells were harvested with TrypLE Express (Gibco), resuspended in electroporation buffer as previously described [55], and a total of 1 × 10^6^ cells were placed into an electroporation cuvette for H-FIRE treatment. H-FIRE treatment was delivered at an electric field of 1500 V/cm which corresponded to the periphery of the electric field distribution for the in vivo patient tumor ablations. We chose for our cell suspension in vitro treatments to correspond to the periphery of the electric field distribution because this area would not have been affected by any potential thermal damage that may have been induced within the proximity of the electrode probe insertion site. Treated CLAC cells, sham-treated CLAC cells, and untreated CLAC cells were plated in 6-well cell culture plates at 5 × 10^5^ cells/mL and cultured at 37 °C with 5% CO_2_ for 2 h and harvested for downstream analysis. For downstream analysis, cell culture supernatant was collected and analyzed for GM-CSF, IFNγ, IL-2, IL-10, IL-8, MCP-1, and TNF-α with the Milliplex canine chemokine/cytokine magnetic bead assay (Millipore Sigma, Burlington, MA, USA) and total RNA was extracted from harvested cells. Total RNA was extracted using the Directzol Microprep RNA extraction kit (Zymo, Irvine, CA, USA); 400 ng of RNA was transcribed into cDNA with the RT^2^ First Strand cDNA transcription kit (Qiagen, Hilden, Germany), and gene expression analysis was performed using the canine specific Cancer Immunity and Inflammation Crosstalk gene array (Qiagen Hilden, Germany) as described above. For analysis, gene expression changes were calculated relative to untreated cells using standard ∆∆Ct methodology. For these preliminary in vitro immune evaluation experiments, one experimental replicate was conducted with technical duplicates.

#### 2.2.3. Statistical Analysis

Statistical comparisons were computed utilizing ordinary one-way analysis of variance test with tukey test for multiple comparisons. The statistical significance value was established as *p* < 0.05. All analyses were conducted with Graph Pad Prism (v10.2).

## 3. Results

### 3.1. In Vivo H-FIRE Ablation: Intraoperative Delivery

#### 3.1.1. Patient Clinical Data

H-FIRE treatment was delivered intra-operatively to a total of five patients. The patient demographics (breed, gender, age, and weight) and tumor details (tumor type, tumor size, location, and size of H-FIRE ablation volume) are outlined in Table 1. The tumor size measurements are based off of CT imaging (measured by G.D.) and the tumor ablation volumes are based off of gross measurements (measured by S.C.O).

#### 3.1.2. H-FIRE Ablation

H-FIRE was successfully administered to a portion of each patient’s lung tumor via a pair of 18-gauge bipolar electrodes and a cycled pulsing protocol [56] was employed. Each patient successfully received the prescribed pulsing protocol of 400 total bursts (split among four electrode pairs) with an applied voltage of 2250 V without notable adverse events. The time required for electrode insertion into tumors was approximately 1–3 min for each patient, and treatment duration was 9.7 ± 1.8 min. Pulses were delivered via cardiac gating as an added safety precaution in four of five patients and no adverse cardiac or other physiologic events were observed across the cohort. In one patient, sufficient ECG electrode contact could not be maintained, likely due to the positioning of the ECG electrode. In this patient, pulses were delivered without synchronization and no cardiac abnormalities were observed. Mild localized contraction of the abdominal wall was occasionally observed, likely due to the proximity of H-FIRE electrode placement to the abdominal wall. These mild muscle contractions were clinically insignificant. Superficial tumor vasculature became observably more prominent during treatment, beginning almost immediately after the administration of the first electrical pulses. No adverse events related to H-FIRE ablation were noted post-operatively while patients were monitored in the intensive care unit.

### 3.2. In Vivo H-FIRE Ablation: Percutaneous Delivery

#### 3.2.1. Patient Clinical Data

H-FIRE treatment was delivered percutaneously to a total of two patients. The patient demographics (breed, gender, age, and weight) and tumor details (tumor type, tumor size, location, and size of H-FIRE ablation volume) are outlined in Table 2. The tumor size measurements are based off of CT imaging (measured by G.D.) and the tumor ablation volumes are based off of gross measurements (measured by S.C.O).

#### 3.2.2. H-FIRE Ablation

All percutaneous CT-guided H-FIRE ablations conducted as a preliminary proof-of-concept for future larger studies, were well tolerated by all patients (*n* = 2). The CT scans confirmed the appropriate placement of the H-FIRE probe in the targeted primary pulmonary tumor (Figure 3A,B). No adverse cardiac or other physiological events were noted during H-FIRE delivery. No signs of pneumothorax or hemorrhage were observed in the post-H-FIRE CT scans. The target voltage and waveform were delivered without limitation in patient 6. In patient 7, the target voltage of 2250 V could not safely be achieved due to muscle contractions. The operating voltage was lowered to 1000 V and the waveform was adjusted to a 1-2-1 µs to effectively deliver H-FIRE treatment. In both cases, 200 pulses were successfully applied.

### 3.3. Evaluations of H-FIRE Ablation Histopathology in Canine Patients

#### 3.3.1. Gross and Histological Outcomes of Intraoperative H-FIRE Ablation

All patient tumor samples (*n* = 5) were histologically diagnosed as pulmonary adenocarcinomas after a detailed review of sections from untreated tumor regions. This diagnosis was made by the identification of one or more typical pulmonary adenocarcinoma patterns in the sample, including the formation of alveoli-like structures, glands with lumens, papillary projections, and others (Figure 2D). All samples exhibited varying degrees of cell death and inflammation in untreated regions, which is a common finding in pulmonary adenocarcinomas.

Grossly, treated areas were characterized by variably well-demarcated foci of tissue hemorrhage (Figure 2B,C). Gross measurements showed the average diameter of the treated region was 1.6 (± 0.8) cm (Table 1) which corresponded closely to the planned ablation parameters of the largest diameter, which was 1.5 cm. Our microscopic evaluation revealed that treated regions of tumor samples (*n* = 3/5 tumor samples) showed increased amounts of cell degeneration and death in H-FIRE-treated samples when compared to paired untreated tumor samples (Table 3). In these samples, the vast majority of cell death appeared as cellular debris multifocally throughout the tumor as well as within the tumor glands. Areas of cell degeneration were characterized by changes in the staining quality of cells (increased pinkness, Figure 2D–F) and the vacuolation of the cytoplasm. Additionally, treated samples were often characterized by prominent hemorrhage that was not a feature of untreated samples (Figure 2E,F). In addition, sections of tumor treated with H-FIRE also exhibited tumor cell loss with varying amounts of fibrin (Figure 2F). All evaluated histological parameters are summarized in Table 3.

#### 3.3.2. Gross and Histological Outcomes of Percutaneous H-FIRE Ablation

Gross and histologic findings for patients treated percutaneously were generally similar to those treated intra-operatively. Untreated tumor samples were utilized for diagnosis and were both also confirmed to be pulmonary adenocarcinomas. Patient 6 exhibited a well-delineated area of grossly visible hemorrhage at the location of the H-FIRE treatment site. This patient also showed significantly increased amounts of hemorrhage and tumor cell degeneration and cell death in treated regions compared to untreated regions. Patient 7 showed no grossly identifiable differences between treated and untreated tumor tissue. Similarly, no significant hemorrhage or tumor cell death were identified in either the treated or untreated tumor samples.

### 3.4. Evaluation of Cell Death and Immune Cell Infiltration Associated with In Vivo H-FIRE Ablation

#### 3.4.1. Evaluation of Cell Death and Macrophage Phenotype via Immunohistochemistry

To investigate the effects of H-FIRE on cell death induction, we qualitatively assessed the expression of activated caspase-3 in tumor samples collected from patients that received intraoperative H-FIRE. In a limited number of tumor samples, we observed increased positive caspase-3 staining in treated regions of tumors compared to paired untreated regions (*n* = 2 of 5) (Figure 4, Table 3). In the remaining three of five patient samples, this trend was not observed (Table 3).

Treated and untreated regions of tumor samples were labeled with macrophage markers, IBA-1, CD206, and iNOS, to investigate macrophage infiltration. Upon examination, the staining for two of the patient samples was deemed suboptimal for adequate evaluation. For the remaining samples deemed adequate for evaluation (*n* = 3), populations of IBA-1 + CD206+ macrophages within untreated and treated tumor samples were most prominent within the connective tissue stroma and sporadically within the glands (structures) formed by tumor cells. We observed populations of IBA-1 + CD206+ macrophages in the untreated and treated regions of tumors, but no additional differences between paired samples (Figure 5A,B, Table 3). We did not observe any IBA-1 + iNOS + macrophages in any collected sample.

#### 3.4.2. Changes in Immune and Inflammatory Genetic Signatures in Intra-Operatively Treated Tumor Samples

Differential gene expression results revealed five genes (*IDO1*, *IL6*, *TNF*, *CD209*, and *FOXP3*) that were, on average, upregulated ≥ 2-fold or downregulated ≤ –2-fold in treated tumor regions relative to untreated tumor regions. The downregulated genes were *IDO1* and *IL-6* and upregulated genes were *TNF*, *CD209,* and *FOXP3* (Figure 6A). Functional gene enrichment analysis associated the five genes with the gene ontology biological pathways related to cell adhesion, leukocyte activation, and the regulation of inflammatory and immune responses (Figure 6B). Differential gene expression patterns differed between each patient and full differential gene expression patterns can be found in Appendix A.

### 3.5. In Vitro Studies

#### 3.5.1. Analysis of H-FIRE Lesion Ablation Size and Lethal Threshold in CLAC Cell Collagen Scaffolds

It was found that the burst delivery rates of 45, 60, or 90 bpm which corresponded to in vivo patient treatment burst delivery rates had a minimal effect on ablation lesion area at 24 h (Table 4, Figure 7A). A moderate burst delivery rate of 60 bursts per minute (bpm) produced a slightly larger ablation lesion area (*p* = 0.025) (Figure 7A) and lethal electric field (*p* = 0.022) (Figure 7B) compared to 45 bpm, but no other significant pairwise differences were observed. We observed no significant differences in caspase-3 staining amongst the evaluated parameters in the collagen scaffold experiments.

After determining lethal electric fields required to elicit cell death 24 h post H-FIRE treatment with our in vitro model, a computational approach was used to (1) estimate the full effects of H-FIRE treatment if the tumors remained in situ for 24 h and (2) analyze the predicted temperature rise due to pulse delivery and estimate the region of thermal damage within each treatment region (Table 5). The results of the computational modeling indicate that patient 4 was predicted to have the largest tumor ablation volume (7.9 cm^3^) and greatest thermal damage volume (5.9 cm^3^) due to its unexpectedly high electrical conductivity, while patient 5 was computed to have the smallest tumor ablation volume (4.1 cm^3^) and smallest thermal damage volume (0.05 cm^3^) due to its low conductivity. The remaining three patients were estimated to have similar predicted ablation volumes and thermal damage volumes (Table 5). The tissue conductivity properties and development of the numerical model have previously been reported [50]. The thermal damage was predicted utilizing the previously described method [57].

#### 3.5.2. In Vitro Immune Evaluation of CLAC Cell Suspensions Post H-FIRE Treatment

At 2 h post H-FIRE treatment, differential gene expression analysis revealed seven genes that were downregulated and six upregulated genes in the treated CLAC cells compared to untreated and sham-treated cells. The downregulated genes *IL15*, *CSF2*, *IL10*, *IL1R2*, *APLNR*, *CXCR4*, and *MYC*, are generally associated with cytokine activity and more specifically, the JAK-STAT signaling pathway (g:Profiler) (Appendix A). The upregulated genes, *CCL2*, *CD244*, *LOC490630*, *CXCR3*, *CCR9*, and *ACKR3,* are generally associated with chemokines and chemokine receptors (Appendix A). We analyzed cell culture supernatant for the immune signaling molecules, GM-CSF, IFNγ, IL-2, IL-10, IL-8, MCP-1, and TNF-α; we observed detectable but insignificant changes in the concentration of the pro-inflammatory chemoattractant IL-8 at 2 h post H-FIRE treatment. The IL-8 concentration was greater in untreated CLAC cell supernatant compared to H-FIRE-treated CLAC cells (Appendix A).

## 4. Discussion

The high prevalence of cancer amongst companion animals promotes the need for advancement in treatment options to improve treatment outcomes and prognosis. The similarities shared between human and companion animal cancer patients allows for animals to serve as comparative models to inform the advancement of treatment options for animal and human cancer patients as well. The tissue ablation modality, H-FIRE, has successfully been employed in the veterinary clinic for the treatment of superficial tumors [40,41,42]. The efficacy and feasibility of H-FIRE treatment for canine hepatocellular carcinomas have also been reported, where clinical adverse events such as muscle contractions and asynchronous cardiac events were not reported to have occurred [35]. Furthermore, canine brain tumors have also been successfully ablated with H-FIRE [39]. The current study was the first in vivo investigation to evaluate the feasibility, safety, and efficacy of in vivo H-FIRE ablation for canine primary lung tumors. We established that H-FIRE ablation can effectively, safely, and precisely be delivered intra-operatively and percutaneously, to client-owned canines with spontaneously occurring primary lung tumors.

Our treat-and-resect study design allowed us to administer H-FIRE treatment to a planned portion of the primary tumor to facilitate an evaluation of treated and untreated tumor regions. During the intraoperative H-FIRE treatment procedure, no major clinical adverse events were observed, and all patients (*n* = 5) recovered as expected. No cardiac arrhythmias were observed, including the one patient in which cardiac gating was not able to be used during the delivery of intraoperative H-FIRE treatment. While no major adverse events were observed, we did observe occasional mild muscle contractions localized to the abdominal wall with certain electrode pairings; these contractions were clinically insignificant. We hypothesized that the observed muscle contractions were likely a result of the proximity of the specific electrode pair to the abdominal wall or the tissue composition of the region of the tumor directly surrounding the electrode pair [34]. However, further investigation with larger patient populations is required to determine the cause and the routine likelihood of abdominal wall contractions, and the potential impact of dog size on the occurrence of the contractions. Importantly, no adverse events were noted in the two patients that received CT-guided percutaneous H-FIRE tumor ablation, supporting the continued development of H-FIRE as a minimally invasive tumor ablation modality for dogs with lung tumors. Of the two patients that we delivered H-FIRE treatment to percutaneously, we were not able to achieve the target treatment voltage of 2250 V for patient 7 and delivered treatment at a voltage of 1000 V. We hypothesize that this may have been due to the tumor’s proximity to the intercostals (ribs) and/or the small size of the dog compared to patient 6. It is worth mentioning that based on the histological outcomes of patient 7′s tumor (the undefined size of the treated tumor region), the modifications made in the treatment parameters may have resulted in an ineffective treatment.

Of the five tumors that received intra-operative H-FIRE treatment, three (patients 1–3) displayed histological differences between the untreated and treated regions of the tumor which were indicative of hemorrhage, cell degeneration, and/or cell death. Amongst these three patients, the degree of histological changes and positive caspase-3 staining varied but was present. Of these three patients, it was estimated that patient 2 had a greater amount of cell death in the treated region of the tumor compared to the untreated region which was evident by increased caspase-3 staining as well as increased tumor cell degeneration and lytic cellular debris compared to the untreated region of the tumor (Table 3). It is unclear why we observed a greater level of histological differences in this patient compared to the other two patients, where we observed similar but more modest changes. Despite the observed similarities in tissue conductive properties and patterns between these three dogs (patients 1–3), which were previously reported [50], we hypothesize that the variation in histological changes was due to tumor tissue composition. This hypothesis is supported by previous studies which report the differences in the electrical resistance of different soft tissue tumors [58,59].

On the contrary, we observed limited differences microscopically between treated and untreated regions of the tumor samples collected from patients 4 and 5 (Table 3). For patient 5, we did not observe significant cellular morphological changes between untreated and treated tumor regions, but grossly we did observe a well demarcated focus of tissue discoloration at the treatment site, and more intense positive caspase-3 staining in the treated region of patient 5′s tumor compared to the untreated region. These findings in patient 5 suggest that H-FIRE did have the desired effect on the targeted tumor region. We hypothesize that cells within the treated tumor region were beginning to undergo apoptosis, but at this early stage this had not manifested into cellular morphological changes and was not evident on H&E staining. Additionally, patient 5′s tumor consisted of foci of osseous metaplasia which were only observed in patient 5′s tumor and potentially could have contributed to the observed histological changes. The foci of osseous metaplasia are structurally dense bone-like tissues which have the potential to impede electrical current [60,61]. Our numerical modeling results predicted that the 24 h post-treatment ablation volume and thermal damage volume would be largest in patient 4 and smallest in patient 5 (Table 5). These predictions are supported by our previously published tissue conductivity data for the intra-operative H-FIRE patients [50].

As previously mentioned, the cell death outcomes associated with H-FIRE are delayed and initial treatment associated cell death occurs over the course of 24 h [32,47,62]. In an effort to predict the tumor cell death outcomes at 24 h post H-FIRE ablation, we investigated the potential of the burst delivery rate to alter tumor cell death. Gaining an understanding of the parameters which may affect tumor cell death post H-FIRE is essential for the clinical application of the treatment modality as the goal of an effective tumor ablation modality is to kill the targeted tumor cells. For the evaluation of burst delivery rate, we utilized an in vitro collagen scaffold method to evaluate three burst delivery rates of 45, 60, and 90 bpm, which paralleled those utilized in vivo for patients. In our collagen scaffold experiments, we observed no significant differences in positive active caspase-3 staining, and only a modest significant difference in the lesion area and lethal threshold between 45 and 60 bpm at 24 h post H-FIRE treatment (Figure 7). These results suggest that the difference in burst delivery rate may not have a significant impact on cell death outcomes at 24 h post delivering H-FIRE treatment. While these results are insightful, it is worth noting that since these experiments were carried out in vitro in 3D collagen scaffolds, we did not directly recapitulate the canine lung tumor microenvironment or inter-patient tumor composition differences. Therefore, to fully understand the effect of burst delivery rate and tumor tissue composition on cell death outcomes post H-FIRE treatment, future clinical studies with larger patient populations are still warranted.

A determining factor of cancer development and progression is the ability of cancer cells to evade inherent immune responses that are in place for detecting and killing cancer cells [63,64,65,66]. Therefore, the ability to stimulate an anti-tumor immune response is considered an essential component of successful cancer therapeutics. Thus, in addition to investigating the ablative outcomes associated with H-FIRE treatment for canine lung tumors, we also evaluated the associated immunological outcomes in vivo and in vitro. Our preliminary in vitro differential gene expression data revealed an upregulation of genes associated with immune cell chemotaxis including the monocyte chemoattractant *CCL2* and the chemokine receptor *CXCR3*, which has been reported to have a dichotomous role in the tumor microenvironment, but is reported to aid in immune cell infiltration which can lead to tumor suppression [67]. The expression of the chemokine receptor *CCR9* has been reported to be associated with improved prognosis in human lung cancer patients [68]. On the contrary, an increased expression of *ACKR3* in lung tumors is associated with an unfavorable prognosis [69]. To provide further insight on the immunomodulatory aspects of H-FIRE treatment, future studies are warranted to evaluate the translation of these genes and subsequent protein expression. At the early timepoint of 2 h post ablation, we observed minimal differences in our multiplex in vivo IHC results for the spatial evaluation of macrophages in the treated and untreated regions of tumor samples. We hypothesized that over time, as the tumor cells targeted by H-FIRE continued to undergo delayed cell death, immune cell infiltration will increase, and further studies are needed to investigate this hypothesis. Our in vivo differential gene expression analysis revealed a downregulation of the gene *IDO1* which codes for a tumor-promoting enzyme [70] and the gene for the metastasis-promoting cytokine *IL6* was also downregulated. *IL6* has been associated with poor prognosis in non-small-cell lung cancer patients [71]. We hypothesize that the upregulation of the gene *CD209* indicates dendritic cell activation [72] leading to the subsequent upregulation of the gene for the pro-inflammatory cytokine *TNF*. Given that the immune tumor microenvironment is typically infiltrated with T-regulatory cells we hypothesize that at the early timepoint of 2 h post H-FIRE treatment, the upregulation of the T-reg transcription factor gene *FOXP3* is likely a result of tumor resident T-regs and not infiltrating T-regs. In previously reported studies, the stimulation of an anti-tumor immune response post H-FIRE is reported to occur at timepoints greater than 24 h post H-FIRE treatment [35,43,47]. At the early time point of 2 h post delivery of H-FIRE treatment, our differential gene expression results are promising and suggest immunomodulation in treated tumor tissue and CLAC cells relative to untreated controls. While both our in vivo and in vitro differential gene expression results suggest immunomodulation post H-FIRE ablation, it is important to acknowledge that the in vitro experiments were conducted with cancer cells only and do not directly mimic the complex TME which we evaluated in our in vivo experiments.

One main component of H-FIRE-induced immunomodulation is hypothesized to be the result of immunogenic cell death (ICD). The stimulation of ICD can lead to the generation of an anti-tumor immune response which is essential for eliminating tumor cells in the H-FIRE-treated tumors and mitigating metastatic disease. The signaling of ICD is driven by the release of immunogenic signaling molecules such as tumor antigens and DAMPs [46,73]. Previous reports have demonstrated the release of core DAMPs such as calreticulin, high mobility group box-1 (HMGB-1), and adenosine triphosphate (ATP) post irreversible electroporation techniques including H-FIRE [32,47,54,55]. In the current pilot study, we did not extensively investigate ICD due to the proof-of-concept study design. However, we did preliminarily investigate the initiation of apoptosis 2 h post H-FIRE treatment in patients and our results suggest that apoptotic cell death may occur post H-FIRE treatment based on cleaved caspase-3 staining, but this was tumor dependent. Our lack of observation of consistent caspase-3 cleavage could be a result of the 2 h post treatment time point, as previous reports have observed caspase-3 cleavage at 6 h [32] and 24 h [30] post treatment. In future studies with larger sample populations, surgical resection at later timepoints, and further investigation into ICD post H-FIRE treatment for pulmonary tumors is warranted. Additionally, mechanistic in vitro and pre-clinical rodent studies would be beneficial for further investigating the cell death, immunological, and ablative outcomes of H- FIRE treatment for pulmonary tumors. Overall, our findings demonstrate the technical feasibility of delivering H-FIRE intra-operatively, and more importantly percutaneously, which will potentially allow for a minimally invasive lung tumor treatment option. Although preliminary, our immune response evaluations suggest that H-FIRE treatment potentially results in the immunomodulation of the local tumor environment which is essential for an effective cancer treatment. Furthermore, in future studies, H-FIRE treatment could be delivered to pulmonary metastatic tumors, a common co-morbidity in cancers such as osteosarcoma [74], for evaluation as novel treatment option.

## 5. Conclusions

In conclusion, this innovative proof-of-concept study demonstrated that H-FIRE can be delivered to pulmonary tumors in canine patients both intra-operatively and percutaneously via CT guidance. We observed no severe clinical adverse events during or post H-FIRE treatment, indicating that the treatment was well tolerated. Our histological evaluation indicated cell death at the treatment site; however, this did vary between patients. The variation in ablation outcomes and subsequent histological evaluations are expected given the heterogenous nature of patient tumor samples but further investigation is warranted. Overall, our exciting and promising findings pave the way for future larger investigations for the treatment of pulmonary tumors with H-FIRE.

## Figures and Tables

**Figure 1 biomedicines-12-02038-f001:**
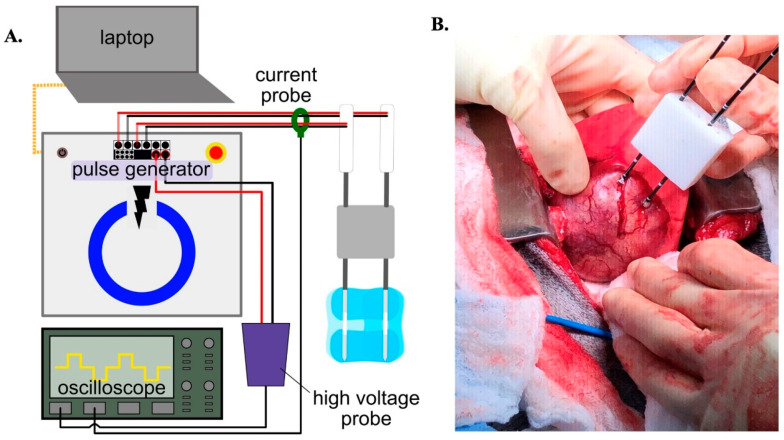
H-FIRE treatment set up and delivery. (**A**) Schematic of H-FIRE equipment. (**B**) Picture depicting the insertion of the bipolar electrodes into a canine lung tumor.

**Figure 2 biomedicines-12-02038-f002:**
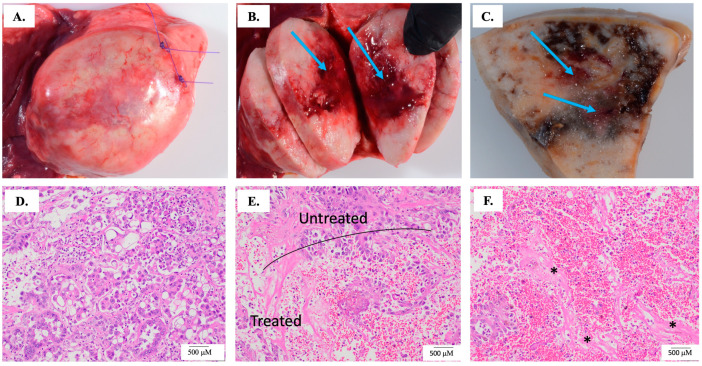
H-FIRE ablation histological outcomes. For all patients untreated and treated tumor samples were evaluated grossly and microscopically. (**A**) Sutures placed at the treated region to ensure positive identification of the ablation site. (**B**) The blue arrows indicate the treated regions of the tumor and (**C**) the blue arrows represent the treated region in a formalin fixed treated sample. (**D**) Representative untreated tumor region. (**E**,**F**) Representative images depicting variable but distinguished amounts of acute hemorrhage, which are the small red circles depicted in the treated region image (**E**) and throughout image (**F**). Prominent acute hemorrhage is not observed in image (**D**) or the untreated region in image (**E**). (**F**) Areas of tumor cell loss and replacement by fibrin are indicated by asterisks.

**Figure 3 biomedicines-12-02038-f003:**
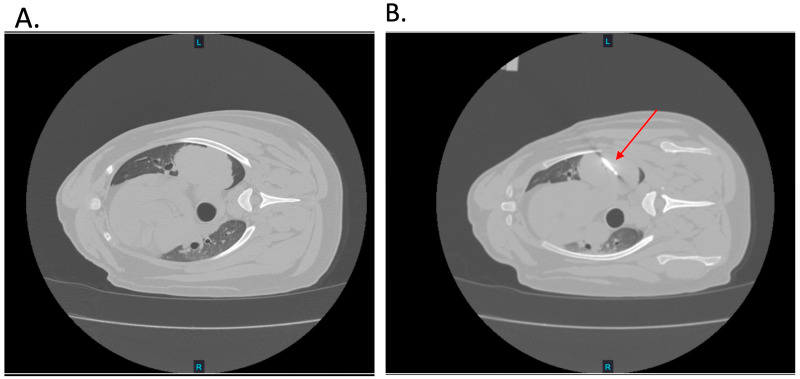
Percutaneous H-FIRE Delivery. (**A**) Representative CT image of patient lung tumor prior to H-FIRE probe insertion. (**B**) Representative CT image depicting the inserted H-FIRE probe into the tumor (red arrow).

**Figure 4 biomedicines-12-02038-f004:**
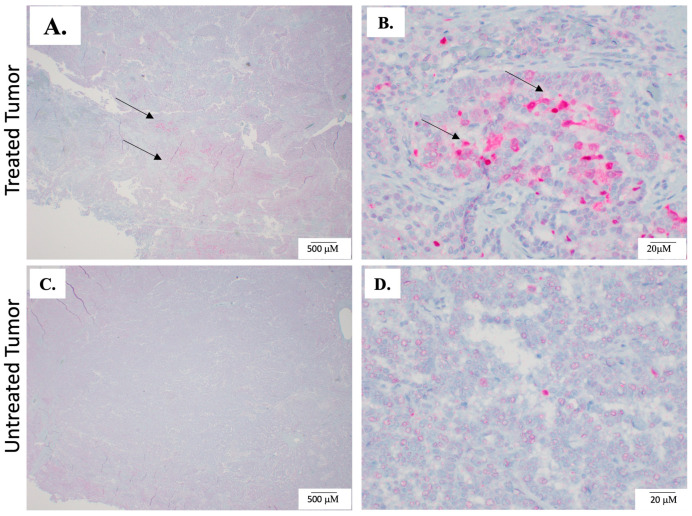
Caspase-3 staining in tumor samples. To investigate the effects of H-FIRE on cell death, paired sections of untreated and treated samples of the tumor were evaluated for the apoptosis marker cleaved caspase-3 with IHC. (**A**,**B**) The dark pink/red staining denoted by arrows is positive cleaved caspase-3 staining and was most prominent in the cytoplasm of treated tumor cells (**B**) In general, treated tumor sections exhibited increased amounts of caspase-3 staining when compared to untreated areas. (**C**,**D**) Representative images of untreated tumor sections, which have minimal cleaved caspase-3 staining.

**Figure 5 biomedicines-12-02038-f005:**
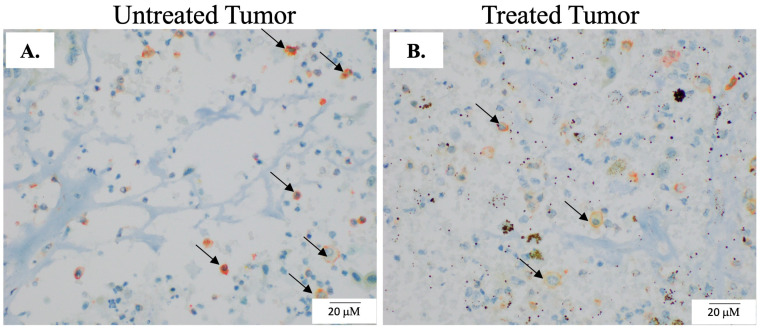
Macrophage infiltration post H-FIRE ablation. To evaluate the phenotype of infiltrating immune cells in paired untreated and treated sections of the tumor, multiplex IHC was performed. Our macrophage panel included IBA-1, CD206, and iNOS. Representative images of paired untreated (**A**) and H-FIRE-treated tumor regions (**B**). The co-staining of IBA-1 and CD206 is depicted by the orange color and arrows.

**Figure 6 biomedicines-12-02038-f006:**
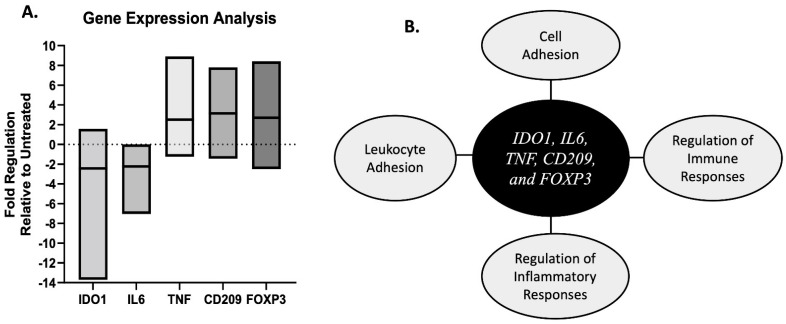
Differential gene expression analysis in paired treated and untreated tumor samples. (**A**) Box and whisker plot of fold regulation values in treated tumor samples relative to untreated tumor samples with a group average fold change of ≥2-fold or ≤−2-fold. (**B**) Diagram of gene ontology biological pathways associated with the five genes with a fold regulation value of ≥2-fold or ≤−2-fold.

**Figure 7 biomedicines-12-02038-f007:**
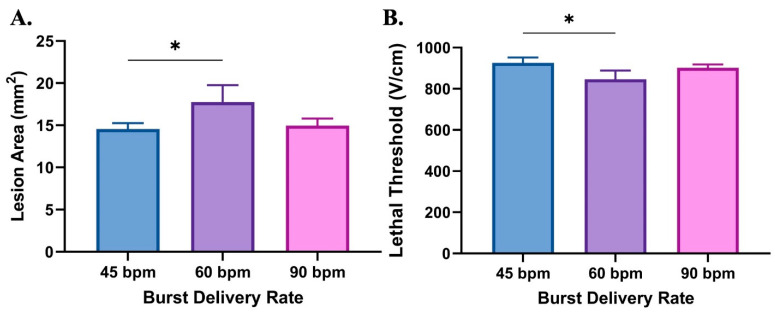
In vitro analysis of H-FIRE ablation lesion size and lethal threshold. (**A**) Ablation lesion area 24 h post delivery of H-FIRE treatment to CLAC hydrogels in vitro. (**B**) Lethal electric field at each burst delivery rate parameter. * *p* < 0.05.

**Table 1 biomedicines-12-02038-t001:** Intra-operative patient demographics.

Patient	Primary Tumor Type and Location	Breed	Gender	Age (yrs)	Weight (kg)	Tumor Size (cm)	Size of Treated Tumor Region (cm)
Patient 1	Adenocarcinoma Right Caudal Lobe	Labrador Retriever	MN	12	42	6.5 × 4.9	1.3 × 1.0 × 1.5
Patient 2	Adenocarcinoma Left Caudal Lobe	Labrador Retriever	MN	8	33	4.1 × 4.6 × 2.8	3.0 × 1.0 × 2.5
Patient 3	Adenocarcinoma Right Middle Lung Lobe	Pitbull Mix	FS	6	29	5.7 × 4 × 5.5	1.5 × 1.0
Patient 4	Adenocarcinoma Right Caudal Lung Lobe	Shih Tzu	MN	11	7	4.5 × 4.5 × 4.5	1.2 × 1.2 × 1.0
Patient 5	Adenocarcinoma Left Cranial Lung Lobe	Retriever Mix	FS	13	18	6.8 × 4.2 × 5.8	1.2 × 1.0 × 1.0

The table displays tumor type and location (right or left lung lobe), and H-FIRE ablation area. MN = male neutered, FS = female spayed. Tumor size is based on CT measurements and treated tumor region is based on histopathology.

**Table 2 biomedicines-12-02038-t002:** Patient demographics for percutaneous H-FIRE patients.

Patient	Primary Tumor Type and Location	Breed	Gender	Age (yrs)	Weight (kg)	Tumor Size (cm)	Size of Treated Tumor Region (cm)
Patient 6	Adenocarcinoma. Left cranial lung lobe.	Pitbull mix	FS	6	29.2	4.7 × 4.1 × 3.1 cm	3 × 1.2 × 0.8
Patient 7	Pulmonary carcinoma, papillary predominant. Right caudal lung lobe.	Shih Tzu	MN	13	7.5	6.44 × 4.43 × 4.41 cm	Undefined

Table displays tumor type and location (right or left lung lobe), and H-FIRE ablation area for patients treated with H-FIRE percutaneously. MN= male neutered, FS= female spayed. Tumor size is based on CT measurements and treated tumor region is based on histopathology.

**Table 3 biomedicines-12-02038-t003:** Gross and microscopic histological findings.

	Gross Observations	Microscopic Observations		Immunohistochemistry	
		Necrosis	Hemorrhage	Caspase-3	Macrophages
Patient 1	Focus of reddening	Untrt: Limited	Untrt: Limited	Untrt: Limited	Untrt: CD206+
		Trt: Present, similar to untrt	Trt: Present, greater than untrt	Trt: Limited, similar to untrt	Trt: CD206+, similar to untrt
Patient 2	Well demarcated focus of hemorrhage and necrosis	Untrt: Limited lytic debris	Untrt: None	Untrt: Limited	Untrt: CD206+
		Trt: Extensive lytic debris, greater than untrt	Trt: Present, greater than untrt	Trt: Present, greater than untrt	Trt: CD206+, similar to untrt
Patient 3	Poorly demarcated focus of hemorrhage	Untrt: Present, mixed lytic and coagulative	Untrt: Limited	Untrt: Limited	Untrt: CD206+
		Trt: Present, predominantly lytic	Trt: Present, greater than untrt	Trt: Limited, similar to untrt	Trt: CD206+, similar to untrt
Patient 4	Poorly demarcated focus of hemorrhage	Untrt: Nuclear debris present	Untrt: Present	Untrt: Present	Untrt: Poor staining
		Trt: Nuclear debris present, similar to untrt	Trt: Present, similar to untrt	Trt: Present, similar to untrt	Trt: Poor staining
Patient 5	Well demarcated focus of tissue discoloration	Untrt: Limited necrotic debris	Untrt: None	Untrt: Limited	Untrt: Poor staining
		Trt: Limited necrotic debris, similar to untrt	Trt: None	Trt: Present, greater than untrt	Trt: Poor staining

Gross observations: characterization of the H-FIRE treatment site. Microscopic observations: characterization of necrosis and hemorrhage in untreated (untrt) and H-FIRE-treated (trt) tumor regions.

**Table 4 biomedicines-12-02038-t004:** In vitro analysis of H-FIRE lesion size and lethal threshold.

Burst Delivery Rate (Hz)	Lesion Area (mm^2^)	Lethal Threshold (V/cm)	N
45	14.6 ± 0.7	926 ± 26	3
60	17.8 ± 2.0	846 ± 42	6
90	15.0 ± 0.8	802 ± 15	3

The lesion area and lethal threshold for each burst delivery rate are reported. N = experimental replications.

**Table 5 biomedicines-12-02038-t005:** Estimated H-FIRE-treated tumor volumes at 24 h post treatment.

⌀	Ablation Volume (cm^3^)	Ablation Height (cm)	Ablation Width (cm)	Ablation Depth (cm)	TD Volume (cm^3^)
1	5.0647	2.8899	2.4112	1.0354	1.5801
2	5.3763	2.9618	2.334	0.97374	1.1622
3	5.2822	2.9513	2.3067	0.97006	1.1041
4	7.8625	3.0975	2.8627	1.2447	5.8559
5	4.0605	2.8589	2.0341	0.83084	0.05425
all	5.5 ± 1.4	3.0 ± 0.1	2.4 ± 0.3	1.0 ± 0.2	2.0 ± 2.3

Computational numerical modeling was utilized to compute the predicted ablation volumes and level of thermal damage (TD) in the patient tumor samples if they were resected at 24 h post H-FIRE treatment. Further details on these methods can be found in previously reported work [50,57].

## Data Availability

The original contributions presented in the study are included in the article/Appendix A, further inquiries can be directed to the corresponding authors.

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
