# Peer review of "Investigation of High Frequency Irreversible Electroporation for Canine Spontaneous Primary Lung Tumor Ablation"

_biomedicines, 2024, doi:10.3390/biomedicines12092038_

Round 1

Reviewer 1 Report

Comments and Suggestions for Authors

The paper presents the test of H-FIRE protocol in canine tumors. The paper is interesting and well written, but it needs some clarifications.

Some suggestion to improve the paper are here listed:  

Line 190 please, when refer to tissue conductivity refers also the electrode distance.

Line 199 please, add a reference for sigmoidal function as model of conductivity

Line 246 what refer 5 uM?

Line 276

Please check Ct methodology (the two delta before)

Line 309 please, specify the meaning of the acronym bpm.

Line 365 Please, provide electrode distance to evaluate the applied electric field.

Line 400-4071 Authors state the reduced applied voltage, please add information of the electrode distance in order to evaluate the reduction of applied electric field.

Line 421 what is n=3/5? Treated and untreated refer to the same tumor and different regions (treated and untreated region)? Please, clarify

Line 425-430 The referred figure is in Material and method section. Please, check the possibility to move in results section.

Fig. 4 what is the difference in panels A and B and also C and D. The scale? In this case prove indication in figure or in caption.

Table 4 What is N column?

Fig. 7 How lesion area was evaluated?

Line 534 why only IL-8 behavior is described? Why other genes are not considered also they show a variation? Please, justify. Please, describe the sham sample characteristics. Are there gene expression differences between in vivo and in vitro experiments? Please, describe.

Line 585 Authors state that hypothesize a variation due to ‘tumor tissue composition’. Please, support this statement with data of tumor tissue composition. It is well known that fibrous part of the tumor or other types of extra-cellular matrix give a different distribution of the electric field in the treatment area. Please, describe this fact with the support of the literature.

Line 601 Authors state ‘These predictions are supported by our previously published tissue…’ Please, provide data for the presented case in order to support statement.

Line 611 The authors state ‘In our collagen scaffold experiments, we observed no significant differences in positive active caspase-3 staining…’ Please, provide data or images to support the statement in comparison with in vivo experiments. Moreover Fig. 7 is provided for in vitro experiments. Are the same parameters evaluated in in vivo experiments? can the authors add a similar evaluation for in vivo experiments?

Please, provide a description of the Intraoperative Delivery and Percutaneous Delivery in order to understand the differences in the application procedures. E.g. add a figure. In the 2 procedure different protocols were used. Please, resume clearly these 2 used protocols in in vivo. Add also a resume of the protocols used in in vitro experiments. Please, add the electrode distance used in each protocols.  

Authors in the material and method section report the description of the tissue conductivity experiments but the results section state that results are in a previous paper. Please, resume these results the cases reported in this paper. How is the thermal damage evaluated? (lines 518-19).

Authors reports in the material and method section also a computation experiments with COMSOL, as treatment planning evaluation. Please, provide model description, solved problem and results of simulations. Were these results are used in this paper? Please, check that all genes evaluated in results were described in materials and methods. Please, provide the used primer sequences.  

Please, check that all data used in discussion are reported and well described in results section (e.g. gene analysis in in vitro and in vivo conditions, COMSOL simulations and conductivity data).

Please, provide information related to ethical request related to patient experiments and data

Author Response

Reviewer 1

  1. Line 190 please, when refer to tissue conductivity refers also the electrode distance.

Response: Thank you for the comment, but could the reviewer please provide more details about their request for the electrode distance. The authors are unclear if the reviewer is requesting more information about the tumor insertion distance or location, spacing of the probes, etc.

  1. Line 199 please, add a reference for sigmoidal function as model of conductivity

Response: Thank you, these references can be found in line 203.

  1. Line 246 what refer 5 uM?

Response: Thank you, the 5 uM refers to the thickness of FFPE tumor tissue scrolls.

  1. Line 276 Please check Ct methodology (the two delta before)

Response: Thank you, this is written correctly, and is a standard methodology.

  1. Line 309 please, specify the meaning of the acronym bpm.

Response: Thank you, this has been addressed.

  1. Line 365 Please, provide electrode distance to evaluate the applied electric field.

Response: Thank you for the comment, but could the reviewer please provide more details about their request for the electrode distance. Additionally, are they requesting that the authors report additional details about the electric field distribution? The request is not clear to the authors.

  1. Line 400-4071 Authors state the reduced applied voltage, please add information of the electrode distance in order to evaluate the reduction of applied electric field.

Response: Thank you, please see above responses regarding similar requests.

  1. Line 421 what is n=3/5? Treated and untreated refer to the same tumor and different regions (treated and untreated region)? Please, clarify

Response: Thank you, additional text has been added to clarify (line 421).

  1. Line 425-430 The referred figure is in Material and method section. Please, check the possibility to move in results section.

Response: Thank you for the suggestion. This figure is referenced in both the materials and methods and results, thus the authors think that that the publishers have opted to position this figure in the M&M location.

  1. Fig. 4 what is the difference in panels A and B and also C and D. The scale? In this case prove indication in figure or in caption.

Response: Thank you for the comment. The authors are not clear what the reviewer is requesting as the difference in the panels is indicated in the figure caption and also on the figure itself. Scale bars are present but have been enlarged during revisions. If the reviewer could provide additional details for their request we are happy to address them.

  1. Table 4 What is N column?

Response: Thank you, this has been clarified in the manuscript.

  1. Fig. 7 How lesion area was evaluated?

Response: Thank you for the question. This information is provided in the materials and methods, please refer to section 2.2.2.3 H-FIRE treatment and viability assessment of collagen scaffolds. If additional details are desired please provide specific requests.

  1. Line 534 why only IL-8 behavior is described? Why other genes are not considered also they show a variation? Please, justify. Please, describe the sham sample characteristics. Are there gene expression differences between in vivo and in vitro experiments? Please, describe.

Response: Thank you for the question. The description of the IL-8 behavior described because it was the only evaluated analyte that we observed changes in at the protein level. This description does not refer to gene expression changes. The authors apologize for the confusion and have evaluated the manuscript and figure text to be sure this is clearly conveyed. The sham treatment information can be found in the materials and methods section of the manuscript. Further information on the gene expression results can be found in the manuscript discussion (lines 625-662).

  1. Line 585 Authors state that hypothesize a variation due to ‘tumor tissue composition’. Please, support this statement with data of tumor tissue composition. It is well known that fibrous part of the tumor or other types of extra-cellular matrix give a different distribution of the electric field in the treatment area. Please, describe this fact with the support of the literature.

Response: Thank you for this comment. The authors acknowledge this difference and have added content to the manuscript discussion (587-589).

  1. Line 601 Authors state ‘These predictions are supported by our previously published tissue…’ Please, provide data for the presented case in order to support statement.

Response: Thank you for this suggestion but since the tissue conductivity data for the 5 intra-operative cases have previously been published the authors feel that it is most appropriate to not report this work again. If the reviewer has additional requests we are happy to consider them.

  1. Line 611 The authors state ‘In our collagen scaffold experiments, we observed no significant differences in positive active caspase-3 staining…’ Please, provide data or images to support the statement in comparison with in vivo experiments. Moreover Fig. 7 is provided for in vitro experiments. Are the same parameters evaluated in in vivo experiments? can the authors add a similar evaluation for in vivo experiments?

Response: Thank you for this suggestion. We have clarified that data for the in vitro collagen scaffold caspase-3 staining is not shown (lines 502-3). The parameters for the in-vitro experiments are reported in the materials and methods. Regarding the experiments reported in Figure 7, a similar evaluation for the in vivo experiments has previously been reported.

  1. Please, provide a description of the Intraoperative Delivery and Percutaneous Delivery in order to understand the differences in the application procedures. E.g. add a figure. In the 2 procedure different protocols were used. Please, resume clearly these 2 used protocols in in vivo. Add also a resume of the protocols used in in vitro experiments. Please, add the electrode distance used in each protocols.  

Response: Thank you for the comment. Please refer to materials and methods section Delivery of H-FIRE Ablation (section 2.1.3) for the description of the H-FIRE delivery procedures and protocols used in vivo, and figures 1 and 3. The materials and methods section 2.2.3 describes the in vitro H-FIRE protocol. If these sections do not appropriately address your concerns we are happy to elaborate further to address more details.

  1. Authors in the material and method section report the description of the tissue conductivity experiments but the results section state that results are in a previous paper. Please, resume these results the cases reported in this paper. How is the thermal damage evaluated? (lines 518-19).

Response: Thank you for your comment and question. To clarify it is not within the goals of this manuscript to describe the tissue conductivity results as these results have previously been reported by our group as referenced in the manuscript. Please refer to this reference for more details on the tissue conductivity results. The details of the thermal evaluation have been previously reported, and this has been clarified in this manuscript (line 524-525).

  1. Authors reports in the material and method section also a computation experiments with COMSOL, as treatment planning evaluation. Please, provide model description, solved problem and results of simulations. Were these results are used in this paper?

Response: Thank you for this suggestion but it is outside the scope of this manuscript to report these results. We believe the reviewer can find these results in our previously published work which is mentioned in lines 137-138. We thank the reviewer for their interest in the Comsol results.

  1. Please, check that all genes evaluated in results were described in materials and methods. Please, provide the used primer sequences.  

Response: Thank you for the comment. All genes are described in supplementary table I.

  1. Please, check that all data used in discussion are reported and well described in results section (e.g. gene analysis in in vitro and in vivo conditions, COMSOL simulations and conductivity data).

Response: Thank you for the comment. We belive that all data that has not previously been reported is described in both the results and discussion. If the reviewer thinks otherwise we would appreciate further details to enable us to further address this concern.

  1. Please, provide information related to ethical request related to patient experiments and data.

Response: Thank you for this request. As described in the materials and methods section Patient Selection, the clinical trial was conducted under an approved IACUC and owner consent was obtained upon enrollment. If the provided information does not adequately address the reviewer’s concerns we are happy to consider adding additional content.

Reviewer 2 Report

Comments and Suggestions for Authors

The article titled “Investigation of high frequency irreversible electroporation for canine spontaneous primary lung tumor ablation” by Hay, Aycock etc. is a well written article summarizing a series of in vivo and in vitro study of H-FIRE. In particular, the investigation on the 5 cases of intraoperative H-FIRE treatment to pulmonary tumors and 2 cases of CT-guided percutaneous H-FIRE ablation presents an important step towards clinical application.

While the reviewer is impressed by the work, there are a few minor concerns and suggestions for the authors, in no particular order.

1. Quantitative presentation of histopathological outcomes.

The results in section “3.3. Evaluations of H-FIRE ablation histopathology in canine patients.” Including table 3 are well written. However, most of the results are presented in a descriptive manner by wordings such as “majority”, “often”, “significantly increased” and “no significant”.

The reviewer would recommend using graphic presentation, such as bar plots, to quantitatively represent the difference between treated and untreated tumors. For instance, percentage of area labeled as necrosis, density of macrophage. This helps the reader better apprehend the results much faster.

2. Evaluation of adverse events.

In the section of 2.1.6. Adverse Event Reporting, it states that any adverse events associated with H-FIRE lung tumor ablation were graded using the VCOG-CTCAEv2. The reviewers did not find gradings of any adverse events following H-FIRE.

3. The process behind estimating in vivo H-FIRE treated tumor volumes, based on results from In Vitro Studies as shown in Table 5, was not clear.

The reviewer is familiar with the subject, however for readers not aware too much of the treatment planning, brief descriptions and references in the section of 2.1.2. Treatment planning can help.  

4. More details on patient 7.

In the article, descriptions such as “we were not able to achieve the target treatment voltage of 2,250V for patient 7 and delivered treatment at a voltage of 1000V” and “In patient 7, the target voltage of 2250V could not safely be achieved due to muscle contractions. The operating voltage was lowered to 1000 V and the waveform was adjusted to a 1-2-1 μs to effectively deliver H-FIRE treatment.” may concern some people if one of the goals of this article is to evaluate the “the technical feasibility of delivering H-FIRE percutaneously under CT guidance”.

There are only 2 cases of CT-guided percutaneous H-FIRE ablation. How can the ineffective treatment (Undefined Size of Treated Tumor Region (cm)) be avoided can be really helpful. 

5. Assessment of macrophage.

Tumor macrophage infiltration typically takes hours to days. However, in this study, tumor resection was performed at least 2 hours after H-FIRE ablation of the tumors.

The reviewer is wondering if the macrophages are recruited to the treated tumor following H-FIRE or were already existing prior to the treatment.

6. Definition of “Treated Tumor Region”

In section 2.2. In Vitro Study, it states that “treat-and-resect nature of the canine clinical study, definitive conclusions regarding the size of the complete ablated volumes were not possible”. The reviewer understands that “treated tumor region is based on histopathology.”

A more definitive description of how the boundary of lesion was determined would be helpful.

7. Table 4 and figure 7.

The reviewer would assume that lower Lethal Threshold will result in larger Lesion Area, if the electric field distribution remains unchanged. However, the results seem not following this assumption. Can the authors clarify my uncertainty?

8. Immune evaluation of CLAC cells can be skipped.

In this article, results are presented only in supplemental materials. There is no comprehensive comparison made between in vivo canine tumor and intro cell suspension.  Treatment conditions are also different (spatially varying electric field vs constant field as shown in Supplement Figure 1 C).

This would include the following sections.

2.2.3. Immune evaluation of CLAC cells treated in vitro with H-FIRE

3.5.2. In vitro immune evaluation of CLAC cell suspensions post H-FIRE treatment

Author Response

  1. Quantitative presentation of histopathological outcomes.

The results in section “3.3. Evaluations of H-FIRE ablation histopathology in canine patients.” Including table 3 are well written. However, most of the results are presented in a descriptive manner by wordings such as “majority”, “often”, “significantly increased” and “no significant”.

The reviewer would recommend using graphic presentation, such as bar plots, to quantitatively represent the difference between treated and untreated tumors. For instance, percentage of area labeled as necrosis, density of macrophage. This helps the reader better apprehend the results much faster.

Response: The authors thank the reviewer for this insightful suggestion. However, since the histological evaluation was only conducted in a semi-quantitative descriptive manner the authors think the way data is currently presented in the manuscript is appropriate. If the reviewer has additional suggestions to improve the readability and/or clarity of the current table and figures the authors will happily consider them.

  1. Evaluation of adverse events.

In the section of 2.1.6. Adverse Event Reporting, it states that any adverse events associated with H-FIRE lung tumor ablation were graded using the VCOG-CTCAEv2. The reviewers did not find gradings of any adverse events following H-FIRE.

Response: Thank you for this comment. The authors did not report full grading because no clinical adverse events were observed during the studies. We did observe mild muscle contractions, but this was not clinically significant (please see results: H-FIRE ablation). We hope that this explanation clarifies the reviewer’s concerns.

  1. The process behind estimating in vivo H-FIRE treated tumor volumes, based on results from In Vitro Studies as shown in Table 5, was not clear.

The reviewer is familiar with the subject, however for readers not aware too much of the treatment planning, brief descriptions and references in the section of 2.1.2. Treatment planning can help.  

 Response: Thank you for this comment. We have added the relevant references to the table 5 legend. However, the authors are not entirely clear about the request from the reviewer. Is the reviewer requesting that we add additional details about the methods for Table 5 as opposed to providing references? If we have not appropriately addressed the reviewers concerns we are happy to address the concern further.

  1. More details on patient 7.

In the article, descriptions such as “we were not able to achieve the target treatment voltage of 2,250V for patient 7 and delivered treatment at a voltage of 1000V” and “In patient 7, the target voltage of 2250V could not safely be achieved due to muscle contractions. The operating voltage was lowered to 1000 V and the waveform was adjusted to a 1-2-1 μs to effectively deliver H-FIRE treatment.” may concern some people if one of the goals of this article is to evaluate the “the technical feasibility of delivering H-FIRE percutaneously under CT guidance”.

There are only 2 cases of CT-guided percutaneous H-FIRE ablation. How can the ineffective treatment (Undefined Size of Treated Tumor Region (cm)) be avoided can be really helpful. 

Response: Thank you for this insightful comment and valid point. As the reviewer mention our overall goal was to demonstrate technical feasibility and safety of delivering H-FIRE percutaneously under CT guidance. However, we do acknowledge that our sample population was small, and that while patient 7’s treatment resulted in no clinical adverse events the changes in the treatment protocol may have resulted in an ineffective treatment. This has been added to the discussion (lines 577-79).

  1. Assessment of macrophage.

Tumor macrophage infiltration typically takes hours to days. However, in this study, tumor resection was performed at least 2 hours after H-FIRE ablation of the tumors.

The reviewer is wondering if the macrophages are recruited to the treated tumor following H-FIRE or were already existing prior to the treatment.

 Response: Thank you, for this comment. It is accurate that macrophage infiltration occurs over the course of typically 2 days. We believe that observed population of macrophages were a composition of infiltrating macrophages and resident macrophages. However, the main takeaway from the macrophage evaluation was the differences we observed in paired treated and untreated tumor regions.

  1. Definition of “Treated Tumor Region”

In section 2.2. In Vitro Study, it states that “treat-and-resect nature of the canine clinical study, definitive conclusions regarding the size of the complete ablated volumes were not possible”. The reviewer understands that “treated tumor region is based on histopathology.”

A more definitive description of how the boundary of lesion was determined would be helpful.

Response: Thank you for the comment. If the boundary of the lesion was not evident histologically the untreated tumor region was selected based on the region furthest away from the probe insertion site. This region also corresponded to a distance greater than treatment volume measurements. We hope this response provides clarity to the reviewer.

  1. Table 4 and figure 7.

The reviewer would assume that lower Lethal Threshold will result in larger Lesion Area, if the electric field distribution remains unchanged. However, the results seem not following this assumption. Can the authors clarify my uncertainty?

 Response: Thank you for the question. This evaluation was conducted to evaluate bust delivery patterns and whether they altered lethal threshold and/or lesion area. For these experiments we utilized an in-vitro collagen scaffold set up and observed no differences. However, the simple nature of the collagen scaffold does not fully recapitulate the tissue environment of a tumor. Thus, our results are only suggestive of the effect of burst delivery patterns. We hope this response clarifies the reviewer’s uncertainty. We are happy to address further questions if needed.

  1. Immune evaluation of CLAC cells can be skipped.

In this article, results are presented only in supplemental materials. There is no comprehensive comparison made between in vivo canine tumor and intro cell suspension.  Treatment conditions are also different (spatially varying electric field vs constant field as shown in Supplement Figure 1 C).

This would include the following sections.

2.2.3. Immune evaluation of CLAC cells treated in vitro with H-FIRE

3.5.2. In vitro immune evaluation of CLAC cell suspensions post H-FIRE treatment

Response: Thank you for this suggestion. While the authors agree that the information provided by the in vitro cell suspension experiments is limited and cannot be directly compared to the in vivo results, we do think the results provide insight. We believe these results can provide the foundation for future larger in vitro studies, thus; we think it is appropriate to include them.

Round 2

Reviewer 1 Report

Comments and Suggestions for Authors

The paper was improved.

Only few suggestions

In figure 4 and 5 express correctly scale bar label with 500 um (use micron symbol) and not 500 uM. 

Please, clarify the electrode distance used in experiments. Please, provide more information about electrode geometry (e.g. needle distance).

Please, add ethical statement at the end of the paper. reference to authorized ethical protocol.

Author Response

Comment 1: In figure 4 and 5 express correctly scale bar label with 500 um (use micron symbol) and not 500 uM. 

Response: Thank you, this has been addressed. 

Comment 2: Please, clarify the electrode distance used in experiments. Please, provide more information about electrode geometry (e.g. needle distance).

Response: Thank you for this comment. Additional information about the electrodes has been added to the manuscript (lines 146-149).

Comment 3: Please, add ethical statement at the end of the paper. reference to authorized ethical protocol.

Response: Thank you. We have added an IRB and informed consent statements. We think these statements will address the request for an ethical statement. Please see lines 734-737.